# CRMS/CFSPID Subjects Carrying D1152H CFTR Variant: Can the Second Variant Be a Predictor of Disease Development?

**DOI:** 10.3390/diagnostics10121080

**Published:** 2020-12-12

**Authors:** Vito Terlizzi, Rita Padoan, Laura Claut, Carla Colombo, Benedetta Fabrizzi, Marco Lucarelli, Sabina Maria Bruno, Alice Castaldo, Paolo Bonomi, Giovanni Taccetti, Antonella Tosco

**Affiliations:** 1Cystic Fibrosis Regional Reference Center, Department of Paediatric Medicine, Anna Meyer Children’s University, 50139 Florence, Italy; giovanni.taccetti@meyer.it; 2Cystic Fibrosis Regional Support Center, Department of Pediatrics, University of Brescia, ASST Spedali Civili Brescia, 25123 Brescia, Italy; ritaf54@gmail.com; 3Cystic Fibrosis Regional Reference Center, Fondazione IRCCS Ca’ Granda Ospedale Maggiore Policlinico, Department of Pathophysiology and Transplantation, University of Milan, 20122 Milan, Italy; laura.claut@policlinico.mi.it (L.C.); carla.colombo@unimi.it (C.C.); 4Cystic Fibrosis Regional Reference Center, Mother-Child Department, United Hospitals, 60126 Ancona, Italy; benedetta.fabrizzi@ospedaliriuniti.marche.it; 5Department of Experimental Medicine, Sapienza University of Rome, 00185 Rome, Italy; marco.lucarelli@uniroma1.it (M.L.); sabinamaria.bruno@gmail.com (S.M.B.); 6Pasteur Institute Cenci Bolognetti Foundation, 00161 Rome, Italy; 7Cystic Fibrosis Regional Reference Center, Paediatric Unit, Department of Translational Medical Sciences, University of Naples Federico II, 80131 Naples, Italy; ali.castaldo@hotmail.com (A.C.); antonellatosco@gmail.com (A.T.); 8Freelance Statistician, 20122 Milan, Italy; paolo_bonomi@outlook.it

**Keywords:** CFTR-RD, CF, IRT, *Pseudomonas aeruginosa*, sweat chloride

## Abstract

Background: There are no predictive factors of evolution of cystic fibrosis (CF) screen positive inconclusive diagnosis subjects (CFSPIDs). Aim: to define the role of the second *CFTR* variant as a predictive factor of disease evolution in CFSPIDs carrying the D1152H variant. Methods: We retrospectively evaluated clinical characteristics and outcome of CFSPIDs carrying the D1152H variant followed at five Italian CF centers. CFSPIDs were divided in two groups: Group A: compound heterozygous for D1152H and a CF-causing variant; Group B: compound heterozygous for D1152H and a: (i) non CF-causing variant, (ii) variant with varying clinical consequences, or (iii) variant with unknown significance. The variants were classified according to CFTR2 mutation database. Results: We enrolled 43 CFSPIDs with at least one D1152H variant: 28 (65.1%) were classified in the group A, and 15 (34.9%) in the Group B. CFSPIDs of group A had the first IRT significantly higher compared to those of group B (*p* < 0.05) and had a more severe clinical outcome during the follow-up. At the end of the study period, after a mean follow-up of 40.6 months (range 6–91.6), 4 (9.3%) out of 43 CFSPIDs progressed to CFTR-RD or CF. All these subjects were in the group A. Conclusions: The genetic profile could help predict the risk of disease evolution in CFSPIDs carrying D1152H, revealing the subjects that need a more frequent follow-up.

## 1. Introduction

Cystic fibrosis (CF) is the most frequent life-limiting autosomal recessive disorder among Caucasians. It is due to mutations in the CF transmembrane conductance regulator (*CFTR*) gene [1]. Due to the severity of the disease, newborn screening (NBS) has been widely implemented internationally [2,3]. A positive immuno-reactive trypsinogen (IRT) at NBS is followed by chloride sweat test and by molecular analysis of *CFTR* mutations. In recent years, the problem of managing subjects with positive IRT and discordant values of sweat chloride (SC) or mutation analysis has become increasingly important for the CF clinicians [4]. These infants are classified as CFTR-related metabolic syndrome (CRMS) in the North-American nomenclature [5] and CF screen positive, inconclusive diagnosis (CFSPID) in Europe [6]. Recently, the CF Foundation consensus guidelines recommended that the two terms had to be harmonized and their use had to become interchangeable to improve international communications and analysis of clinical outcomes [7,8]. Infants are currently classified as CRMS/CFSPID if they have a positive IRT test plus: (i) SC <30 mmol/L and 2 CF causing variants of which at least 1 with unclear phenotypic consequences, or (ii) SC 30–59 mmol/L and 1 or 0 CF-causing variants [6].

Many of these children will remain asymptomatic, but later in life, a percentage of them will develop symptoms suggestive of CFTR-related disorder (CFTR-RD) or CF [9,10,11,12]. The prevalence of CRMS/CFSPID cases is extremely variable in different countries, depending on the specific NBS algorithm and on the type of genetic analysis carried out [13,14]. Similarly, the percentage of cases that evolve to CF or to CFTR-RD and the age of evolution are variable in turn, and no markers are available so far to predict the cases of CRMS/CFSPID with a higher risk to evolve [12].

This poses a severe dilemma on how to monitor CFSPID patients, with what intensity, with what protocols, and for how long, given that only a small percentage of them will evolve.

The D1152H (p.Asp1152His, c.3454G>C) is a class IV CFTR variant causing the production of an abnormal CFTR protein that is only partially activated by cAMP, consequently it is associated to some CFTR protein residual activity [15]. Currently, the CFTR2 database classifies D1152H as a variant with varying clinical consequences (VVCC), i.e., the D1152H *in trans* with a CF-causing variant may or may not cause CF (http://www.http.com//www.cftr2). The clinical expression of CF in patients with the D1152H variant is heterogeneous, and seem to depend mainly on the causing variant in trans: more severe in individuals carrying a class I–II–III pathogenetic variant on the other allele than in those with class IV–V pathogenetic variant or homozygous for D1152H [15,16].

The Italian CF population carries less frequently the F508del mutation as compared to the North-American and northern European populations; on the contrary, CF pathogenetic variants with residual function or VVCC are more frequent, such as D1152H present in 2.9% of CF patients [16].

The D1152H mutation has been frequently observed in subjects with CRMS/CFSPID [12]. Thus, we retrospectively evaluated clinical characteristics and outcome of CRMS/CFSPID infants carrying the D1152H pathogenetic variant, followed at five Italian reference CF centers in order to define the role of the second CFTR pathogenetic variant as a predictive factor of disease evolution.

## 2. Materials and Methods

The study was approved by the Ethical committee of the CF coordinator centre (Florence, Ethics Clearance number 140/2018, on 1 October 2018) and subsequently by all Ethical committees of the participating CF centers.

### 2.1. Diagnostic Test and Clinical Data

All CRMS/CFSPID subjects born from 1 January 2011 to 31 August 2018, followed at 5 CF Italian centers and carrying at least a D1152H variant on one allele, were selected. The retrospective study ended on 31 December 2018.

CFTR gene sequencing [17] and multiplex ligation-dependent probe amplification (MLPA), were performed in all CRMS/CFSPID with SC in the intermediate range and for one center [12], also in the presence of one pathogenic variant after first-level genetic analysis regardless of SC value.

Clinical, genetic and microbiological data, SC levels [18], and informations about management during follow-up (growth, pulmonary exacerbation, complications) were collected. All clinical data were recorded by research assistants (monitors) trained in audit visits.

Pancreatic sufficiency was based on fecal elastase value higher than 200 µg/g [19]. Oropharyngeal swab cultures were undertaken according to the clinical practice of each center. Chronic infection was defined following Leeds criteria [20]. Pulmonary exacerbations were defined according to the CF Foundation criteria [21].

According to the genetic profile and to the CFTR2 mutation database (https://cftr2.org/), CRMS/CFSPID subjects were divided in two groups:Group A: compound heterozygous for D1152H and a CF causing variant;Group B: compound heterozygous for D1152H and a: (i) non-CF causing variant; (ii) variant with varying clinical consequences; (iii) variant of unknown significance.

During follow-up, CRMS/CFSPID subjects were reclassified by clinicians as: (i) CF, if with pathological SC (≥60 mmol/L) or multi-organ involvement [22]; CFTR-related disorder (CFTR-RD), in case of SC in the intermediate or normal range and single-organ involvement [23].

The study was funded by the Italian Cystic Fibrosis Research Foundation (FFC#30/2018) and approved by the Ethical committee of the CF coordinator center (Florence, Ethics Clearance number 140/2018) and subsequently by all Ethical committees of the participating CF centers.

### 2.2. Statistical Analysis

Descriptive statistics for quantitative variables were obtained according to normal distribution tests. Comparisons between independent samples were performed using Levene’s test to assess the equality of variances and two-tailed Student’s t-test for the equality of the means.

Chi-Squared test was used to determine independence of two categorical variables. For small samples a second check was performed with The Fisher Exact.

The level of statistical significance was expressed as a p-value, and it was considered statistically significant if it was less than 0.05.

## 3. Results

We enrolled 43 CFSPID subjects (21 males, 48.8%) with at least one D1152H variant. Among the 43 subjects enrolled, 28 (65.1%) out of 43 were included in group A; 15 (34.9%) out of 43 in group B (Appendix A). The list of *CFTR* variants of the two groups is shown in Appendix A. All mothers of the subjects were Caucasian. All the subjects were pancreatic sufficient at birth and at the end of the study period (31 December 2018).

As shown in Table 1 and in Figure 1, CRMS/CFSPID subjects of group A showed a mean first blood IRT significantly higher compared to subjects of group B (96.6 ng/mL vs. 78.4 ng/mL; *p* = 0.007), while there was no significant difference between the first SC value within the two groups (27.8 mmol/L vs. 24.2 mmol/L *p* = 0.328).

The clinical characteristics of CRMS/CFSPID subjects of two groups are summarized in Table 2.

During follow-up, CRMS/CFSPID subjects of group A had episodes of pancreatitis in 3 (11.5%) and metabolic alkalosis in 1 (3.8%) out of 26 cases, while none of the subjects in group B had symptoms. Furthermore, subjects of group A underwent chest x-ray and physiotherapy treatment significantly more frequently than subjects of group B. In the same way, *P. aeruginosa* was more frequently isolated from CRMS/CFSPID subjects of group A (38.5% vs. 7.7%; *p* < 0.05). None of these children developed a chronic infection (Table 3).

At the end of the study period, after a mean follow up of 40.6 months (range 6–91.6), 39 (88.6%) out of 43 asymptomatic children remained in the CRMS/CFSPID category; on the other hand, 3 CRMS/CFSPID subjects progressed to CFTR-RD diagnosis for pancreatitis and one converted to CF for SC in pathological range. All these subjects were in the group A (Table 3).

## 4. Discussion

This is the first study to analyze the role of the *CFTR* genotype as a predictor of disease evolution in CRMS/CFSPID children. The study demonstrates that CRMS/CFSPID subjects with D1152H in trans with a *CFTR* variant classified as CF-causing according to the CFTR2 database (Group A) more frequently develop episodes of pancreatitis, isolation of *P. aeruginosa* and respiratory exacerbations in the first year of life, or they required respiratory physiotherapy, radiological examinations, or saline supplementation. Furthermore, these subjects have a higher risk to evolve to CF.

Selecting the cases of CRMS/CFSPID that need a follow-up is an important objective in NBS programs, since: (i) The number of such cases identified during NBS is dramatically increasing [9,10,11,12], the referring and following of these infants in CF centers creates a burden on both their families and on the health-care system [24], and finally, (ii) the evaluation of such subjects in CF center enhances the risk of bacterial colonizations [25].

Thus, since a balance is needed between over medicalization and under-treatment of CRMS/CFSPID subjects, the possibility to stratify their risk to progress is mandatory. At present, there are no predictive markers of evolution of CRMS/CFSPID subjects in CF or in CFTR-RD during follow-up [12] and it is difficult for pediatricians to reassure the families in regards to their kid’s future [12,24].

As we demonstrated for the risk of pancreatitis in CF children [26], it would be very useful also in CRMS/CFSPID subjects to be able to predict, already on the basis of the individual genetic profile, the risk of developing CF related symptoms in order to diversify the follow up.

Few studies have looked for predictors of disease evolution in CRMS/CFSPID: A study suggested that NBS IRT levels at birth may help predict the likelihood of CF among infants with CRMS/CFSPID [27]. In our cohort, the group with a causing variant in trans had a significantly higher IRT value than the other group, but it was not possible to identify a discriminating cut-off value for the large overlap of the values between the two groups. Moreover, the IRT values obtained in the four subjects subsequently evolved were confused within the distribution of the subjects of both the groups, as it appears in Figure 1. On the other hand, in a previous study on the long-term follow-up of subjects with CRMS/CFSPID, we demonstrated that IRT at birth was unable to predict the evolution of CRMS/CFSPID subjects [12].

Recently, a role of lung clearance index to identify lung disease early and thus the risk to evolve to asymptomatic CRMS/CFSPID subjects has been reported [28], but our data did not confirm such evidence.

Other studies suggested that the conversion of CRMS/CFSPID to CF could be predicted by the values of SC at birth [28,29], although other studied did not confirm this evidence: A study on sixty-three CRMS/CFSPID subjects matched with 63 CF subjects diagnosed by NBS showed that those CRMS/CFSPID converted to CF had SC values at birth comparable to those who did not convert [30]. Similarly, in our cohort and in a previous study by our group including eighty-two CRMS/CFSPID infants, the values of SC at birth in infants converted to CF or to CFTR-RD were not significantly different as compared to not-evolved cases.

Finally, no clinical and no anthropometric data collected during the monitoring of CRMS/CFSPID subjects was useful to predict their outcome in the present, and in most previous studies.

To data, serial repeated sweat testing remains the only non-invasive parameter to monitor the disease development in CRMS/CFSPID cohorts [11,31], since the evaluation of CFTR function on the basis of intestinal current measurement in samples from rectal biopsy or nasal potential difference in children are slightly invasive, although reproducible, according to guidelines [32,33,34].

Some studies demonstrated that the extended *CFTR* gene analysis enhances the percentage of CRMS/CFSPID cases diagnosed at birth [11,12,27]. The subsequent follow-up of such subjects permits to identify early cases that evolve to CFTR-RD [12] or to CF [11,12,27] not revealed by NBS. However, sequencing analysis often leads to the identification of (i) non-CF causing variants such as L997F [35,36,37], (ii) variants with varying clinical consequences, and (iii) variants of unknown significance. In the present study, we demonstrated that CRMS/CFSPID subjects carrying the D1152H variant in trans with a mutation classified as caused by CFTR2 had a more severe expression during the follow-up and a higher risk to evolve in CFTR-RD and CF.

In our previous Italian study on patients heterozygous for D1152H and carrying a class I-III or IV–V variant in trans (*N* = 84), only 1 patient (4%) in the group of 25 patients < 10 y carrying a severe class I-III CFTR variant (i.e., who would belong to group A in the current study) developed chronic or recurrent pancreatitis [15]. In this cohort of CRMS/CFSPID infants, however, as many as 3 out of 26 patients in group A (11.5%) aged < 8 years showed episodes of pancreatitis and were classified as CFTR-RD. This discrepancy may be due to the current increased attention and awareness of the risk of pancreatitis in children carrying a D1152H and a CF causing variant.

We studied CRMS/CFSPID infants with at least one copy of the D1152H, because: (i) Such variant has a high frequency in the Italian CF population [16] and in CRMS/CFSPID subjects [11,12], and (ii) previous data indicated that the clinical expression of CF in patients with the D1152H is heterogeneous and may in part depend on the pathogenic variant in trans [15]. Furthermore, recent data show that, over time, variants with residual function such as D1152H can determine a markedly reduced life expectancy [38,39] and the Food and Drug Administration approved the use of ivacaftor, a CFTR potentiator, to also treat patients carrying D1152H [40]. Thus, the identification of CRMS/CFSPID infants at higher risk to evolve in CF and the early diagnosis of the progression could be a useful tool also for the early identification of subjects eligible for the use of CFTR modulators.

## 5. Conclusions

Our data show that most CRMS/CFSPID subjects carrying the D1152H variant remain without a definitive diagnosis after several years of follow-up. However, a minority of them develops CF-related symptoms, and this occurs more frequently in individuals with a CF-causing variant in trans. They also develop clinical complications more frequently during the follow-up. Such data, which must be confirmed in CRMS/CFSPID subjects with other variants, suggest that the careful analysis of the CFTR genotype of CRMS/CFSPID subjects may help to define the cases with a higher risk to progress to CF/CFTR-RD.

## Figures and Tables

**Figure 1 diagnostics-10-01080-f001:**
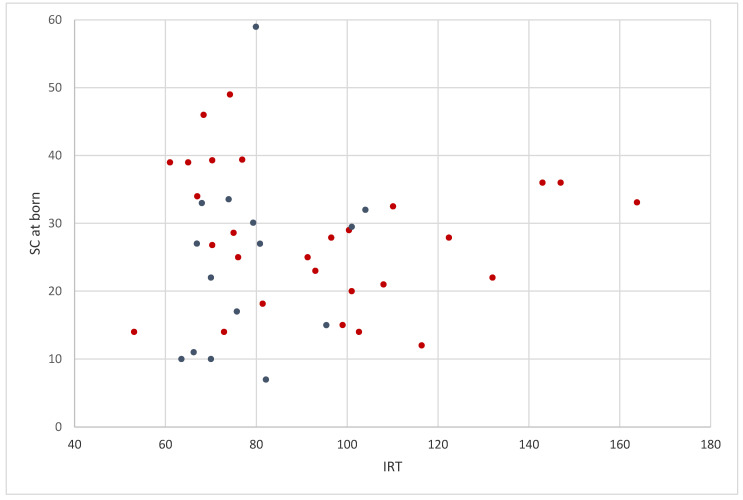
IRT (ng/mL) and first sweat chloride (mmol/L) values of 43 CRMS/CFSPID subjects (in red for infants of group A and in blue for infants of group B).

**Table 1 diagnostics-10-01080-t001:** Immunoreactive trypsinogen (IRT) and sweat chloride value in CRMS/CFSPID subjects.

	*n*	Group A	nr	Group B	*p*
Mean IRT (ng/mL)	28	96.6	15	78.4	0.007
IQR§ 25–75		74.2–110.1		68–82.1	
Mean Sweat chloride (mmol/L)	28	27.8	15	24.2	0.328
IQR 25–75		21–34		11–32	

**Table 2 diagnostics-10-01080-t002:** Clinical characteristics of CRMS/CFSPID subjects.

	Group A			Group B	
	*N*	%	*N*	%	*p*
**A. Clinical symptoms**					
Pancreatic sufficiency	28/28	100%	15/15	100%	
Pancreatitis	3/26	11.5%	0/15	0.0%	0.110
Metabolic hypochloremic alkalosis	1/26	3.8%	0/15	0.0%	0.442
**B. Respiratory Infection**					
Pseudomonas aeruginosa *	10/26	38.5	1/13	7.7%	0.044
MSSA	12/26 °	46.2%	6/13	46.2%	0.871
MRSA	3/26 ˆ	11.5%	2/13	15.4%	0.735
**C. Pulmonary exacerbations**					
N of antibiotic therapy in the first year	25	1.8	13	0.6	0.027
IQR 25–75		0–3		0–0	
**D. Diagnostic tools performed**					
Oropharyngeal swab	26/26	100.0%	13/15	86.7%	0.056
Chest X-ray	25/26	96.2%	7/15	46.7%	<0.001
**E. Therapies**					
Chest physiotherapy	18/26	69.2%	2/15	13.3%	0.001
Saline Supplementation	15/26	57.7%	3/15	20.0%	0.019

Abbreviations: MSSA: methicillin-susceptible Staphylococcus aureus; MRSA: methicillin-resistent Staphylococcus aureus. * No subject developed chronic Pseudomonas aeruginosa infection. ° Two subjects developed chronic MSSA infection. ˆ One subject developed chronic MRSA infection.

**Table 3 diagnostics-10-01080-t003:** Characteristics of CRMS/CSPID subjects who progressed to CFTR-RD/CF.

	Age of Final Diagnosis (Months)	Final Diagnosis	First Variant	Second Variant	Symptoms	First sweat Chloride (mmol/L)	Last Sweat Chloride (mmol/L)
1	43	CF	D1152H	G542X	None	34	71
2	7	CFTR-RD	D1152H	R1158X	Pancreatitis	20	20
3	27	CFTR-RD	D1152H	L732X	Pancreatitis	21	21
4	11	CFTR-RD	D1152H	F508del	Pancreatitis	12	24

Abbreviations: CF: Cystic fibrosis; CFTR-RD: Cystic fibrosis transmembrane conductance regulator Related Disorder.

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
