# Peer review of "CRMS/CFSPID Subjects Carrying D1152H CFTR Variant: Can the Second Variant Be a Predictor of Disease Development?"

_diagnostics, 2020, doi:10.3390/diagnostics10121080_

Round 1
Reviewer 1 Report
This is a very interesting study investigating an important issue of the clinical outcome of the subjects heterozygous for D1152H variant depending on the second mutation.
There are some minor concerns to be clarified
Table 2
There were 15 subjects in group the B. However the authors wrote, that 12/12 patients had sufficient pancreas. What happened with the other 3?
Results:
The authors stated, than no patient developed chronic P. aeruginosa infection. It would be also interesting to know how many of those infected with S. aureus developed chronic infection.
Author Response
Manuscript ID
diagnostics-1030398
Reviewer 1
This is a very interesting study investigating an important issue of the clinical outcome of the subjects heterozygous for D1152H variant depending on the second mutation.
There are some minor concerns to be clarified
Table 2
There were 15 subjects in group the B. However the authors wrote, that 12/12 patients had sufficient pancreas. What happened with the other 3?
Re: This is a mistake. I corrected the sentence.
Results:
The authors stated, than no patient developed chronic P. aeruginosa infection. It would be also interesting to know how many of those infected with S. aureus developed chronic infection.
Re: We have added the requested data in the table.
Reviewer 2 Report
This retrospective clinical study extends and replenishes a previous study on the phenotype of individuals carrying the class IV D1152H CFTR mutation on at least one allele (ref. 15) by showing in a rather large cohort of Italian CFSPID patients (43; classified by NBS) that the CFTR variant on the other allele (CF-causing, group A; non CF-causing/VVCC/unknown significance: group B) is a main predictor of the risk for disease development, i.e. CFTR-RD (pancreatitis) and CF (group A>>group B). Stratification of the risk of disease development in CRMS/CFSPID infants is a key objective in NBS because it can help to prevent over-medicalization or under-treatment. Moreover it may facilitate the decision to start early treatment with a CFTR modulator, such as ivacaftor. Therefore this study is both timely and of significant clinical relevance.
Specific comments/questions:
- In the previous, even larger study of Italian patients heterozygous for D1152H and carrying a class I-III or IV-V mutation in trans (N=84), only 1 patient (4%) in the group of 25 patients < 10y carrying a severe class I-III CFTR mutation (i.e. who would belong to group A in the current study) developed chronic or recurrent pancreatitis (ref. 15, Table 2). In the current CFSPID study, however, as many as 4 out of 26 patients in group A (15.4%) showed episodes of pancreatitis and were classified as CFTR-RD (Table 2 and 3). Considering that all patients enrolled in the CFSPID study were <8 y old (born between January 2011 and December 2018), the authors should clarify this apparent discrepancy.
- Why are only 3 out of the 4 patients with signs of pancreatitis in group A diagnosed as CFTR-RD? What is the second variant in case of the pancreatitis patient not classified as CFTR-RD, i.e. not included in Table 3?
- An earlier publication (ref. 26) reported that a low PIP score (<0.25) could be used to predict the risk of acute pancreatitis in CF children, and that the PIP score correlated with the sweat chloride test. Could the authors also calculate the PIP score for the present cohort of CFSPID individuals and was it lower in the 4 patients with episodes of pancreatitis of whom 3/4 have normal sweat chloride (Table 3)?
- 210-213: Here the authors mention serial repeated sweat testing as the only non-invasive parameter to monitor disease development in CFSPID cohorts, and disqualify ICM and NPD in children as invasive and poorly reproducible. However they seem to ignore the fact that all 3 patients who progressed to CFTR-RD failed to develop elevated sweat chloride values during this transition (Table 3). Furthermore, it is true that ICM is slightly invasive (though painless and without any serious complication), but it is certainly not poorly reproducible if carried out according to the guidelines of the ECFS-ICM SOP (cf. Minso R et al 2020 BMJOpen Respir Res: e000736). Therefore the authors are advised to tone down their statement.
- The manuscript contains a number of typo’s and inaccuracies that need further correction. For example: l. 172: causing accordingly=CF-causing according; l. 184: reassure=to reassure; l. 196: IRT at born=at birth; l. 324: Cystic Fibrosis Foundation (in duplicate); l. 339: resoultion=resolution; l. 342: encironment=environment.
Author Response
Reviewer 2
This retrospective clinical study extends and replenishes a previous study on the phenotype of individuals carrying the class IV D1152H CFTR mutation on at least one allele (ref. 15) by showing in a rather large cohort of Italian CFSPID patients (43; classified by NBS) that the CFTR variant on the other allele (CF-causing, group A; non CF-causing/VVCC/unknown significance: group B) is a main predictor of the risk for disease development, i.e. CFTR-RD (pancreatitis) and CF (group A>>group B). Stratification of the risk of disease development in CRMS/CFSPID infants is a key objective in NBS because it can help to prevent over-medicalization or under-treatment. Moreover it may facilitate the decision to start early treatment with a CFTR modulator, such as ivacaftor. Therefore this study is both timely and of significant clinical relevance.
Specific comments/questions:
- In the previous, even larger study of Italian patients heterozygous for D1152H and carrying a class I-III or IV-V mutation in trans (N=84), only 1 patient (4%) in the group of 25 patients < 10y carrying a severe class I-III CFTR mutation (i.e. who would belong to group A in the current study) developed chronic or recurrent pancreatitis (ref. 15, Table 2). In the current CFSPID study, however, as many as 4 out of 26 patients in group A (15.4%) showed episodes of pancreatitis and were classified as CFTR-RD (Table 2 and 3). Considering that all patients enrolled in the CFSPID study were <8 y old (born between January 2011 and December 2018), the authors should clarify this apparent discrepancy.
Re: We thank the reviewer, it is an interesting consideration. Most likely there is currently a greater attention to the risk of pancreatitis in subjects with mutations with variable clinical consequences, especially such as D1152H. I think several episodes of pancreatitis were previously unrecognized.
- Why are only 3 out of the 4 patients with signs of pancreatitis in group A diagnosed as CFTR-RD? What is the second variant in case of the pancreatitis patient not classified as CFTR-RD, i.e. not included in Table 3?
Re: the correct number is 3. The data has been corrected in the table and in the text.
- An earlier publication (ref. 26) reported that a low PIP score (<0.25) could be used to predict the risk of acute pancreatitis in CF children, and that the PIP score correlated with the sweat chloride test. Could the authors also calculate the PIP score for the present cohort of CFSPID individuals and was it lower in the 4 patients with episodes of pancreatitis of whom 3/4 have normal sweat chloride (Table 3)?
Re: The PIP score is a useful predictor of the risk of pancreatitis in CF patients. In this cohort of patients the time of follow up is too short. Moreover, since our population does not have the diagnosis of Cystic Fibrosis with pancreatic insufficiency, the PIP score is not calculable.
- 210-213: Here the authors mention serial repeated sweat testing as the only non-invasive parameter to monitor disease development in CFSPID cohorts, and disqualify ICM and NPD in children as invasive and poorly reproducible. However they seem to ignore the fact that all 3 patients who progressed to CFTR-RD failed to develop elevated sweat chloride values during this transition (Table 3). Furthermore, it is true that ICM is slightly invasive (though painless and without any serious complication), but it is certainly not poorly reproducible if carried out according to the guidelines of the ECFS-ICM SOP (cf. Minso R et al 2020 BMJOpen Respir Res: e000736). Therefore the authors are advised to tone down their statement.
Re: We have modified the sentence as indicated. However, the definition of the 3 subjects as CFTR-RD is precisely based on the non-pathological sweat test and single organ involvement. In these patients, repeat sweat testing is important because of the risk of pathological values over time.
- The manuscript contains a number of typo’s and inaccuracies that need further correction. For example: l. 172: causing accordingly=CF-causing according; l. 184: reassure=to reassure; l. 196: IRT at born=at birth; l. 324: Cystic Fibrosis Foundation (in duplicate); l. 339: resoultion=resolution; l. 342: encironment=environment.
Re: we have correct the mistakes.